# Quantifying *BRCA1* and *BRCA2* mRNA Isoform Expression Levels in Single Cells

**DOI:** 10.3390/ijms20030693

**Published:** 2019-02-06

**Authors:** Vanessa L. Lattimore, John F. Pearson, Arthur E. Morley-Bunker, kConFab Investigators, Amanda B. Spurdle, Bridget A. Robinson, Margaret J. Currie, Logan C. Walker

**Affiliations:** 1Department of Pathology and Biomedical Science, University of Otago, Christchurch 8011, New Zealand; vanessa.lattimore@otago.ac.nz (V.L.L.); john.pearson@otago.ac.nz (J.F.P.); morar032@student.otago.ac.nz (A.E.M.-B.); bridget.robinson@cdhb.health.nz (B.A.R.); margaret.currie@otago.ac.nz (M.J.C.); 2Research Department, Peter MacCallum Cancer Center, Melbourne 3010, Australia; heather.thorne@petermac.org; 3The Sir Peter MacCallum Department of Oncology, University of Melbourne, Parkville 3010, Australia; 4Genetics and Computational Biology Division, QIMR Berghofer Medical Research Institute, Brisbane, Queenslan 4006, Australia; amanda.Spurdle@qimrberghofer.edu.au; 5Canterbury Regional Cancer and Haematology Service, Canterbury District Health Board, Christchurch Hospital, Christchurch 8011, New Zealand

**Keywords:** splicing, single cell, mRNA expression, *BRCA1*, *BRCA2*, RNAscope

## Abstract

*BRCA1* and *BRCA2* spliceogenic variants are often associated with an elevated risk of breast and ovarian cancers. Analyses of *BRCA1* and *BRCA2* splicing patterns have traditionally used technologies that sample a population of cells but do not account for the variation that may be present between individual cells. This novel proof of concept study utilises RNA in situ hybridisation to measure the absolute expression of *BRCA1* and *BRCA2* mRNA splicing events in single lymphoblastoid cells containing known spliceogenic variants (*BRCA1*c.671-2 A>G or *BRCA2*c.7988 A>T). We observed a large proportion of cells (>42%) in each sample that did not express mRNA for the targeted gene. Increased levels (average mRNA molecules per cell) of *BRCA2* ∆17_18 were observed in the cells containing the known spliceogenic variant *BRCA2*c.7988 A>T, but cells containing *BRCA1*c.671-2 A>G were not found to express significantly increased levels of *BRCA1* ∆11, as had been shown previously. Instead, we show for each variant carrier sample that a higher proportion of cells expressed the targeted splicing event compared to control cells. These results indicate that *BRCA1*/*2* mRNA is expressed stochastically, suggesting that previously reported results using RT-PCR may have been influenced by the number of cells with *BRCA1*/*2* mRNA expression and may not represent an elevation of constitutive mRNA expression. Detection of mRNA expression in single cells allows for a more comprehensive understanding of how spliceogenic variants influence the expression of mRNA isoforms. However, further research is required to assess the utility of this technology to measure the expression of predicted spliceogenic *BRCA1* and *BRCA2* variants in a diagnostic setting.

## 1. Introduction

Germline *BRCA1* and *BRCA2* variants can confer an increased risk of breast and ovarian cancers, and many of these variants are known to disrupt mRNA splicing, as seen in Figure 1. *BRCA1* and *BRCA2* mRNA splicing patterns have been explored in depth using a number of advanced sequencing technologies to highlight significant diversity, both in the number and level of the alternative events expressed naturally by these genes [1,2,3]. This diversity can be attributed to the detection methods, diurnal rhythmic patterns and random variation, and/or the effects that spliceogenic variants have on the use of regulatory motifs by splicing machinery [4,5].

In particular, individual cells are known to express genes in a stochastic manner, occurring in bursts, irrespective of the transcription factors present [6,7], suggesting that the detected *BRCA1* and *BRCA2* mRNA expression levels may be influenced by such fluctuations. Expression based assays (such as reverse transcriptase-PCR and RNA-seq) measure average mRNA expression in a population of cells. However, the individual cells of the population are likely to be producing transcripts at very different levels, potentially providing an additional measure of mRNA expression patterns associated with spliceogenic variants. Advanced in situ hybridisation (ISH) technologies provide the means to observe actual levels of mRNA transcripts in single cells, in addition to detail on spatial expression within a cell. Measuring gene splicing at the single cell level may provide a more comprehensive method of evaluating the biological and clinical significance of alternative isoforms. In this study, we utilised RNAscope [8], a unique method of RNA in situ hybridisation, to show the effect of known *BRCA1* and *BRCA2* spliceogenic variants on mRNA transcript expression in single cells.

## 2. Results

In this study, RNAscope allowed the detection of specific *BRCA1* and *BRCA2* mRNA splicing events in single cells, from individuals with or without germline spliceogenic variants. Results showed that the proportion of cells expressing *BRCA1* transcripts ranged from 4% (2/50) in the control to 44% (22/50) in the *BRCA1*c.671-2 A>G carrier, as seen in Table 1. By comparison, the proportion of cells expressing *BRCA2* transcripts ranged from 24% (12/50) in the control to 58% (29/50) in the *BRCA2*c.7988 A>T carrier. Results from the negative and positive control probes from the RNAscope Multiplex Fluorescent Detection Kit are shown in the Appendix A. Cell-specific *BRCA1* mRNA transcript expression ranged from 0 to 17 (mean = 2.1) *BRCA1* molecules per cell in the *BRCA1*c.671-2 A>G carrier, and 0 to 10 (mean = 0.3) in the control. The *BRCA2* mRNA transcript expression ranged from 0 to 29 (mean = 3.2) molecules per cell in the *BRCA2*c.7988 A>T carrier, and 0 to 5 (mean = 0.6) in the control as shown in Table 1 and Appendix A. Both carrier and non-carrier lymphoblastoid cell lines (LCL) treated with a nonsense mediated decay (NMD) inhibitor showed an average increase in *BRCA1* and *BRCA2* mRNA molecules per cell compared to non-treated LCLs (fold changes observed: *BRCA1* carrier = 5.2×; *BRCA1* control = 1.2×; *BRCA2* carrier = 1.6×; *BRCA2* control = 4.4×), as seen in Appendix A and the Appendix A. To be consistent with previous publications [4,9], we have focused our analysis on NMD-treated cells.

Previous studies [4,10] conducting RT-PCR and RNA-seq analyses of *BRCA1* mRNA expression in LCLs carrying the spliceogenic variants *BRCA1*c.671-2 A>G and *BRCA2*c.7988 A>T have reported increased expression of the ∆11 and ∆17_18 splicing events, respectively. However, these studies were unable to elucidate *BRCA1* and *BRCA2* expression patterns at the single cell level. Using RNAscope, we show that the *BRCA1*c.671-2 A>G variant carrier LCL showed no significant difference in the total number of detected *BRCA1* ∆11 mRNAs compared to the control LCL across the cells assayed in each sample, demonstrated by *p* = 0.48 in Table 1, Figure 2. However, the number of cells containing the *BRCA1* ∆11 splicing event was shown to be significantly greater in the variant carrier compared to the control (*p* = 0.01, as seen in Table 1). The detection of *BRCA2* ∆17_18 mRNA molecules in *BRCA2*c.7988 A>T variant carrier cells was significantly higher compared to the control (*p* = 0.04, as seen in Table 1 and Figure 2). Furthermore, the number of cells containing the *BRCA2* ∆17_18 splicing event was also shown to be significantly greater in the variant carrier compared to the control LCL (*p* = 0.04, as seen in Table 1).

Together these results show that RNA in situ hybridisation was able to demonstrate the increased expression of *BRCA2* ∆17_18 splicing events previously reported. Increasing cell-sampling size may allow the 1.3-fold increase of *BRCA1* ∆11 detected previously [10] to also be identified. In addition to previous studies, we show that these samples show the punctuated expression of *BRCA1* and *BRCA2*, with a large proportion of cells containing no detectable target mRNA molecules, as seen in Table 1.

## 3. Discussion

mRNA splicing assays typically follow a population-based sampling approach, using sequencing or PCR-based methods to simultaneously detect and quantify mRNA splice isoforms in a large number of cells. Attempts to quantify the observed *BRCA1* and *BRCA2* mRNA expression in normal samples have identified huge variability, both between and within samples [4,10]. This suggests that a shift to focusing on individual cells may help determine the extent to which cellular heterogeneity contributes to this variation.

Here, specific *BRCA1* and *BRCA2* mRNA isoforms were quantified in individual cells. A large proportion of cells in all samples studied were not found to express any of the *BRCA1*/*2* mRNA transcripts. Furthermore, the expression of *BRCA1* ∆11 and *BRCA2* ∆17_18 isoforms were observed in a greater proportion of cells isolated from the carriers of spliceogenic variants, compared to those observed from the control. However, this study has revealed that transcripts were not uniformly expressed across all cells, with a large proportion of cells containing no detectable target mRNA molecules. This suggests that results previously reported using RT-PCR [4] and RNA-seq [3] might reflect the number of cells with gene activity, rather than simply detecting an elevation of constitutive activity. Such inter-cellular variation may account for some of the inter-sample variability previously observed using RT-PCR and RNA-seq [3,4]. These results are also consistent with previous reports of variable gene expression between individual cells [6,11], but have not previously been shown for *BRCA1* and *BRCA2*.

mRNA isoforms detected in samples at very low levels might actually be highly abundant in a small number of cells, but this observation can only be detected at the single-cell level, as seen in Figure 3. Such isoform expression changes may be associated with an increased risk of disease, as a higher proportion of aberrant transcripts in a single cell may significantly increase the likelihood of cellular disruption, leading to cancer development, but may go undetected using cell population-based methods. RNAscope enables *BRCA1* and *BRCA2* alternative mRNA isoforms to be studied at a single molecule resolution to determine the expression range at which each isoform is tolerated in normal cells. Such information may help identify gene expression changes in patient samples that are outside the expected range. RNAscope may offer an alternative means for future diagnostic work to assess splice isoforms across individual cells using archival tissue (e.g., formalin fixed paraffin embedded tissue) and to identify cell-specific overexpression in different tissue types. 

Nonsense-mediated decay inhibitors are often used in splicing analysis to prevent the degradation of mRNA [4], thus improving the chances of detecting the present mRNA transcripts. Our study has showed both an increased number of *BRCA1* and *BRCA2* transcripts, and an increase in the number of cells containing transcripts, in the treated samples compared to the untreated samples. Higher expression in the treated samples facilitated the detection of the targeted alternative splicing event, suggesting that NMD inhibitor treatment is advantageous when analysing *BRCA1* or *BRCA2* transcript expression with RNAscope.

Our study is proof of concept, demonstrating the utility of RNA in situ hybridisation to characterise the effects of spliceogenic variants at the single cell level. Our study had several potential limitations, including the use of a single control and the number of cells analysed (*n* = 50), which might have led to an under-representation of some isoforms, such as those that were lowly expressed. Increasing the total number of cells counted in each sample may provide a more accurate survey of isoform expression patterns. However, counting a greater number of cells would be relatively laborious using manual assessment. This challenge could be alleviated by using image analysis software, such as the Aperio RNA ISH Algorithm (Leica Biosystems), or Cell Profiler (http://cellprofiler.org/). It is important to note that there is paucity in published data relating to the efficacy of these image analysis software tools. It is unclear why some cells contained a greater number of C1 (targeting the deletion of interest) probe signals compared to the reference C2 (targets all gene-specific transcripts) probe signals as seen in Appendix A. While the C1 and C2 regions of any mature mRNA containing both domains are adjacent and thus predicted to generate corresponding signals in overlapping locations within the cell, our current scoring method does not allow for this analysis. Future research may employ confocal microscopy and image analysis tools, which enable the 3D imaging of transcripts detected using dual probe fluorescence. This could be a technical issue relating to probe binding and signal amplification efficiency, which could be addressed by sampling sufficient cell numbers. This disparity may also be due to probes binding to pre-mRNA. However, the time from transcription to pre-mRNA splicing is thought to be very short, only 2.5–3 min [12], while the mRNA half-life of *BRCA1* and *BRCA2* is 4.5 and 4.8 h [13], respectively. The proportion of C1 and C2 signals binding to pre-mRNA for either gene is therefore likely to be very small at any given time. A further explanation of this observation may be in relation to the expression of alternative spliced transcripts, which could be missing large parts of the C2, but not C1, probe-binding region. While predominant alternative isoforms are known for both *BRCA1* (∆3) and *BRCA2* (∆12) [1,2], these exons are each only a relatively small proportion of the possible C2 probe-binding region and so are not expected to influence the detection of the gene transcript. However, additional splicing events that are further restricting the number of binding sites may be present.

The unique probe design RNAscope uses to obtain high specificity also limits the number of isoforms that can be detected with this technique. Three probe pairs are required to return a detectable signal, with each pair taking up 36–50 nucleotides of the target sequence, while 10–20 pairs are recommended to gain optimal detection [8]. This limits the minimal target size to ~350 bases in length, which is larger than many single *BRCA1* and *BRCA2* exons. With the advent of BaseScope^TM^ (Advanced Cell Diagnostics, Newark, CA, USA), specific mRNA transcripts are now able to be detected using shorter probes [14]. However, this method is specific to a targeted splice junction and therefore may not detect multiple isoforms with overlapping deletion events. For example, the detection of *BRCA1* ∆11 using BaseScope^TM^ probes spanning the exon 10_12 junction will not identify the *BRCA1* ∆9_11 splice event, even though both isoforms contain a deletion of the large exon 11 sequence. Thus, assays targeting exon 11 may be utilised to examine potentially ‘leaky’ variants, which give rise to multiple alternative isoforms.

To our knowledge, this is the first study to reveal *BRCA1* and *BRCA2* mRNA expression patterns at the single cell level in LCLs. This technology could potentially play an important role in assessing variable expression levels of *BRCA1* and *BRCA2* isoforms by establishing whether mRNA isoforms detected in samples at very low levels may actually be highly abundant, but in a small number of cells. Such findings may be associated with an increased risk of disease, yet might go undetected using cell population-based methods because the number of cells expressing the aberrant change is likely to be insufficient to indicate an association with disease. Spliceogenic variants previously dismissed as low-risk might need to be re-evaluated at the single cell level to determine if aberrant mRNA isoforms are being expressed at high levels but in a low number of cells. Furthermore, single cell expression analysis provides an important means for investigating the poorly explored link between many human diseases and gene expression variability (variance) [15,16]. Such studies may uncover crucial indicators of early cancer development and would open the door for technologies like RNAscope to be applied in diagnostic work in the future.

## 4. Materials and Methods

### 4.1. Study Samples

Four lymphoblastoid cell lines (LCLs), two derived from rare variant carriers (*BRCA1*c.671-2 A>G and *BRCA2*c.7988 A>T) and two from healthy controls, were obtained from the Kathleen Cuningham Foundation Consortium for Research into Familial Breast Cancer (kConFab; Melbourne, Victoria, Australia; http://www.kconfab.org/Index.shtml). The variant carriers were selected for analysis as they contained variants known to disrupt normal splicing patterns [4]. Cell lines were cultured in RPMI 1640 media, with 10% fetal calf serum and 1% Penicillin Streptomycin, while being incubated at 37 °C in a 5% CO_2_ atmosphere. mRNA was extracted from LCLs cultured with and without the nonsense mediated decay (NMD) inhibitor cycloheximide (100 µg/mL) for 4 h.

### 4.2. Probe Design

Labelled probes were designed to detect splicing alterations known to be upregulated in the variant carriers, namely, *BRCA1* ∆11 upregulated in *BRCA1*c.671-2 A>G and *BRCA2* ∆17_18 upregulated in *BRCA2*c.7988 A>T variant carriers [4,9]. Probe design is detailed in Appendix A. Briefly, two sets of probes (C1_green_ and C2_red_) were designed for *BRCA1* and *BRCA2*. C1_green_ probes were designed to target the deleted mRNA region (*BRCA1* exon 11 or *BRCA2* exons 17_18). C2_red_ probes were designed to target a large transcript region, with the pretext that at least part of that region is represented in all expressed transcripts as seen in Figure 2; panels A and D. Total mRNA expression in each cell was determined from the number of instances where both C1_green_ and C2_red_ probe signals are detected, whereas the expression of each of the targeted alternative splicing events of interest was calculated by subtracting the number C1 signals from the number of C2 signals in each cell, as seen in Figure 2, panels A and D. Control probes for housekeeping gene *UBC* (positive control) tissue and bacterial gene *dapB* (negative control) were included for each assay.

### 4.3. RNAscope Assay

RNA in situ hybridisation was performed once for each sample using the RNAscope^®^ Fluorescent Multiplex kit (Advanced Cell Diagnostics, Newark, CA, USA) according to the manufacturer’s instructions specified in RNAscope^®^ Fluorescent Assay for PBMC and Non-Adherent Cells (Part 1), and the RNAscope^®^ Fluorescent Multiplex Kit User Manual Part 2 [8] (see Appendix A).

### 4.4. Slide Imaging

Fluorescent signals were captured and manually quantified using an epifluorescent Zeiss AxioVision microscope and associated software (AxioVersion 4.5. Apotome software, Carl Zeiss Microscopy, LLC, Thornwood, New York, NY, USA). The EC plan-Neofluar 40×/1.3 oil Dic M27 objective was used with DAPI, TRITC and FITC filters. Fluorescent signals were counted independently for each wavelength in 50 non-overlapping cells. Accurate counting was conducted by scanning through different focal planes in the *z*-*axis* for each cell. Each signal reportedly corresponds to one RNA molecule [8].

### 4.5. Data Acquisition and Statistical Analysis

The number of transcripts containing the targeted deletion event in each cell was determined through differences in the number of fluorescently labelled probes detected for each colour channel. For example, where there is an equal number of *BRCA1* C1_green_ (exon 11) and C2_red_ (exons 1–10) fluorescent signals in a cell, we determine that all detected transcripts contain exon 11 as seen in Appendix A. An increased ratio of C2_red_ to C1_green_ signals indicates an exon-skipping event (∆11), as shown in Appendix A. The Yates corrected chi-square test was used to test for independence in the presence of the delta events between cases and controls. This is a conservative test used to prevent an overestimation of significance when analysing small datasets, especially when some values in the test are below the recommended minimum of 10.

## Figures and Tables

**Figure 1 ijms-20-00693-f001:**
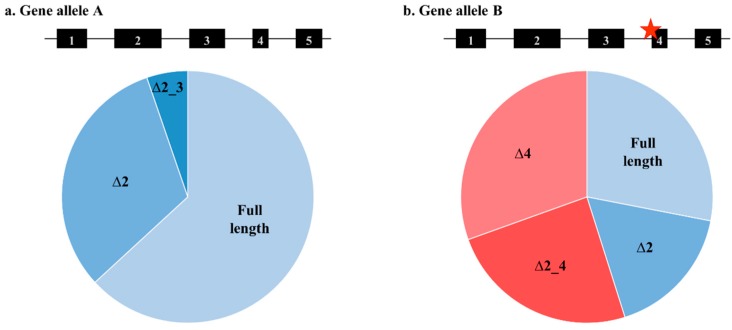
Schematic highlighting potential mRNA splicing isoform expression changes with and without a splice-disrupting genetic variant present. (**a**) The total expression of normally spliced mRNA isoforms. (**b**) mRNA isoform expression with a spliceogenic variant present. Variant location is indicated by the star.

**Figure 2 ijms-20-00693-f002:**
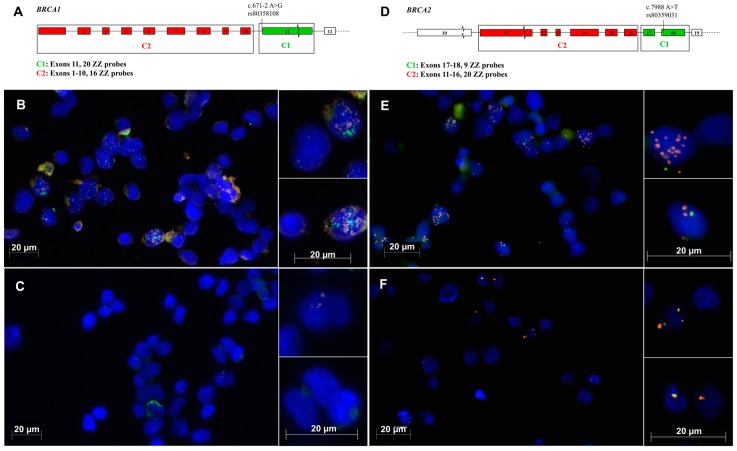
*BRCA1* and *BRCA2* mRNA expression detected by RNAscope. Location and number of RNAscope fluorescent probes used to specifically detect *BRCA1* ∆11 and total *BRCA1* mRNA (**A**). Colour represents the detectable fluorescence of the probes. *BRCA1* mRNA expression levels in individual cells of a lymphoblastoid cell line (LCL) containing *BRCA1*c. 671-2 A>G (**B**) and in a control LCL (**C**). Location and number of RNAscope fluorescent probes used to specifically detect *BRCA2* ∆17-18 and total *BRCA2* mRNA (**D**). Colour represents the detectable fluorescence of the probes. *BRCA2* mRNA levels in individual cells of an LCL containing *BRCA2*c.7988 A>T (**E**), and in a control LCL (**F**). Green signals (C1_green_ probes) indicate the detection of the targeted deleted mRNA region (*BRCA1* exon 11 or *BRCA2* exons 17_18). Red signals (C2_red_ probes) indicate the detection of all expressed transcripts from the targeted gene. Samples were treated with the NMD inhibitor cycloheximide.

**Figure 3 ijms-20-00693-f003:**
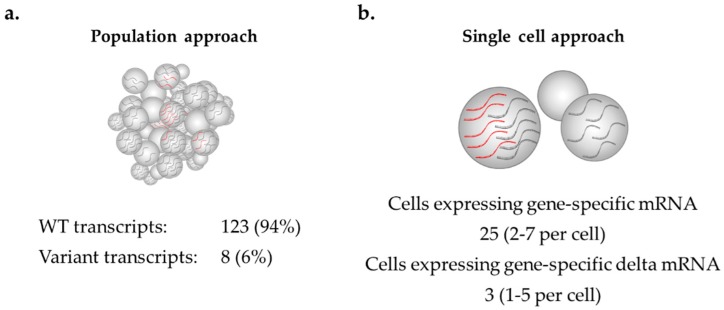
Exemplar of possible observations when measuring mRNA expression of normally expressed and aberrant transcripts in the same cells using a population approach (**a**) or a single cell approach (**b**).

**Table 1 ijms-20-00693-t001:** Quantifying ∆11 and ∆17_18 splicing events in LCL variant carriers (*BRCA1* c.671-2 A>G and *BRCA2* c.7988 A>T, respectively) and controls.

	Number of Cells Expressing Transcripts (*n* = 50)	Total Number of Transcripts ^a^ (Range Per Cell)	Total Number of ΔEvents ^b^ (Range Per Cell)	*p*-Value ^c^ (OR; CI)	Number of Cells with a ΔEvent	Number of Cells Without a ΔEvent	*p*-Value ^c^ (OR; CI)
*BRCA1* c.671-2 A>G	22 (44%)	104 (0–17)	26 (0–8)	0.48 (0.53; 0.16–1.78)	8	37	0.01 (10.60; 2.26–137.60)
Control	2 (4%)	13 (0–10)	5 (0–5)		1	49	
*BRCA2* c.7988 A>T	29 (58%)	160 (0–29)	71 (0–21)	0.04 (2.74; 1.11–6.71)	14	36	0.04 (3.5; 1.15–10.63)
Control	12 (24%)	31 (0–5)	7 (0–2)		5	45	

^a^ Number of signals from C1 and C2 probes across 50 non-overlapping cells. ^b^ Number of Δ events = C2-C1 probe signals per cell for 50 non-overlapping cells. ^c^ Yates corrected chi-squared test. Abbreviations: CI: confidence interval; OR: odds ratio.

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
