# Peer review of "Quantifying BRCA1 and BRCA2 mRNA Isoform Expression Levels in Single Cells"

_ijms, 2019, doi:10.3390/ijms20030693_

Reviewer 1 Report

Lattimore et al. use RNA in situ hybridization to analyze the relative abundances of full length and exon-skipping BRCA1/2 mRNA isoforms in lymphoblastoid cell lines containing wild type BRCA1/2 or splice disrupting variants.  This approach addresses the problem of inconsistent measurements of both background levels and mutation-associated levels of alternate splicing events in various cell lines by examining the frequency of mRNA species on an individual cell basis.  Specifically, the authors examine the relative frequencies of BRCA1 exon 11-skipping events in wild type LCLs and those containing BRCA1c.671-2 A>G and BRCA2 exon 17-18 skipping events in wild type LCLs and those containing BRCA2c.7988 A>T.  

Using the RNAscope method of in situ hybridization, these authors saw no significant difference in BRCA1 exon 11-skipping events to full length BRCA1 mRNA ratios between wild type and BRCA1 c.671-2A>G mutation carriers at the population level.  While this result differs from previous reports showing that BRCA1 c.671-2A>G is a spliceogenic mutation (eg Whiley et al., 2014), high levels of variability between experiments is commonly observed.  More importantly, the authors noted that only 4% of wild type cells but 44% of BRCA1 c.671-2A>G mutation carriers showed expression of any BRCA1 mRNA at all.  Moreover, there was a significant difference in the proportion of cells expressing the exon11-skipping isoform in the BRCA1 c.671-2A>G LCLs compared with the wild type LCLs (8/45 vs 1/50).  These results suggest the spliceogenic mutation may have a significant impact on the frequency of an exon-skipping event that may be detected by single cell analysis but not at the level of cell populations.

Likewise, the authors showed a significant difference in BRCA2 exon 17-18-skipping events to full length BRCA2 mRNA ratios between wild type and known spliceogenic BRCA2 c.7988A>T mutation carriers at the population level.  More importantly, however, they show that only 24% of wild type LCLs but 58% of BRCA2 c.7988A>T mutation carriers show expression of any BRCA2 mRNA at all.  Moreover, there was a significant difference in the proportion of cells expressing the exon 17-18-skipping isoform in the BRCA2 c.7988A>T LCLs compared with the wild type LCLs (14/50 vs 5/50).  Again, these results suggest the spliceogenic mutation may have a significant impact on the frequency of an exon-skipping event that may be detected by single cell analysis but not at the level of cell populations.   

This work is a potentially valuable proof of concept that will allow meaningful interpretations of the spontaneous exon-skipping rates in wild type genes as well as the impact of spliceogenic mutations.  Single cell analysis has the potential to obviate the quantitative confusion created by gene expression occurring in small and variable numbers of cells in a population.  While this study could potentially serve as a template for future studies into alternate splicing events, it is difficult to follow the logic of scoring the red and green signals as two domains of the same molecule as described.  As the experimental method is presented, the C2 (red, Alexa 550) set of probes recognize all BRCA1/2 mRNAs that do not have significant exon-skipping events in the exons covered, and serve as the baseline level of all BRCA1/2 molecules under consideration.  The C1 (green, Alexa 488) set of probes cover the exon(s) that are skipped at some background level and, perhaps, at a higher level in the presence of a spliceogenic mutation.  To determine the rate of exon skipping, one would therefore compare the frequency of signals that show both red and green signals (indicating no exon-skipping event in the region of interest) with those showing only red (indicating an exon-skipping event).  This interpretation is consistent with the description in Table 1 of the total number of delta events as “Number of delta events = C2-C1probe signals per cell for 50 non-overlapping cells,” setting aside the question of whether the green signals spatially coincide with a subset of the red signals.  Yet the exemplar images shown in Supplementary Figure 1 suggest that the red and green signals are being scored as separate molecules, not separate domains of the same molecule.  See the legend statement “A: cellular mRNA that does not carry the targeted deletion as indicated by equal numbers of both green (C1) and red (C2) signals.”  Note the red and green signals are clearly spatially distinct in these images and cannot represent different domains of the same molecule (the emission wavelengths of Alexas 488 and 550 are considerably greater than the distances between the C1 and C2 probes used for both BRCA1 and BRCA2, and would therefore overlap).  If the green signal masks the presence of the red signal coming from the same molecule, then the Sup Fig 1A image would indicate an equal number of exon skipping events and full length mRNAs, not mRNA that fails to carry the targeted deletion as indicated in the legend.  There are fluorescence imaging methods for distinguishing molecules/complexes that carry one fluorescence signal from molecules/complexes that carry both.  Results might resemble the yellow-orange signal seen in the center-left hand side of the lower right panel in Supplementary Figure 3B.

Apart from clarifying the signal-counting method and interpretation, the authors should consider discussing the subcellular localization of the signals.  If alternately spliced molecules contribute to the pool of translatable mRNAs, one would expect them to accumulate in the cytoplasm (see again the yellow-orange signal seen in the center-left hand side of the lower right panel in Supplementary Figure 3B and the upper inset in Figure 2C).  C1 and C2 signals within the nucleus may represent spliced mRNAs, unspliced pre-mRNAs, or arrested intermediates in the splicing/mis-splicing processes.  This is less of a concern for the BRCA1 C2 probe and the BRCA2 C1 probe that are composed of multiple exons that would likely have to be spliced together to generate good signal.  But BRCA1 C1 and BRCA2 C2 probes each contain long exon 11 sequences that might produce a strong signal in RNA molecules that are not fully spliced mRNAs.  Fluorescence images in Supplementary Figure 1 and similar images, where most signals co-inside with nuclear staining, should be described as potentially spliced mRNA variants, but also possibly other types of molecules along the splicing pathway.     

Minor concerns:

1.  Line 80 states that “Results from the negative and positive control probes from the…detection system are shown in Supplementary Figures 2 and 3.”  It would be useful to know more about what is included in these controls.  Do the negative controls lack expressed BRCA1/2 mRNA, or are they simply probe components that lack oligos?

2.  In the Supplementary Information, there are a couple of sentences that read like proprietary kit instructions that need to be spelled out as much as possible.  These include line 36, “Pretreat 3 solution was added…” and line 44 “…was incubated with AMP 1-FL for 30 minutes, followed by AMP 2-FL…”

Author Response

1.    It is difficult to follow the logic of scoring the red and green signals as two domains of the same molecule as described. Cell signals are spatially distinct and so can’t be seen to represent the same molecule.

The reviewer is correct in that the fluorescent signals are spatially distinct and so it is not possible to determine if two signals are from the same molecule. Thus, red and green signals were scored separately by focusing on each wavelength independently, eliminating the possibility that one signal masks another. Signal counts were all undertaken manually with focal plane adjustment to ensure that all of the signals present in the 3-dimentional nature of the cells studied were detected. We have added the following information to the methods section.

Lines 258-260: “Fluorescent signals were counted independently for each wavelength in 50 non-overlapping cells. Accurate counting was conducted by scanning through different focal planes in the z-axis for each cell.”

2.    There are fluorescence imaging methods for distinguishing molecules/complexes that carry one fluorescence signal from molecules/complexes that carry both.  Results might resemble the yellow-orange signal seen in the center-left hand side of the lower right panel in Supplementary Figure 3B.

We agree with the reviewer that automated methods would be ideal for this process. However, suitable tools that had the capability to accurately count signals in a 3D environment were not an option for this study.

3.    The authors should consider discussing the subcellular localization of the signals… BRCA1 C1 and BRCA2 C2 probes each contain long exon 11 sequences that might produce a strong signal in RNA molecules that are not fully spliced mRNAs.

The reviewer raises a valid point regarding the location of the mRNA in relation to each cell nucleus and what RNA molecules may be detected. The RNAscope assay allows for the 3D nature of each cell to be maintained, so signals for mRNA located in the cytoplasm are often overlapping the nucleus in a 2D image. This can make it difficult to distinguish between mRNA and nucleic pre-mRNA, if signals were to bind to the latter. However, the time from transcription to pre-mRNA splicing is thought be minutes whereas the mRNA half-life of BRCA1 and BRCA2 is several hours. The relative proportion of pre-mRNA present for these genes is therefore likely to be very small at any given time. We have added the following explanation for why there is a disparity between the two signals.

Lines 189-193: “This disparity may also be due to probes binding to pre-mRNA. However, the time from transcription to pre-mRNA splicing is thought be very short, only 2.5-3 minutes [12], while the mRNA half-life of BRCA1 and BRCA2 is 4.5 and 4.8 hours [13], respectively. The proportion of C1 and C2 signals binding to pre-mRNA for either gene is therefore likely to be very small at any given time.

4.    Line 80 states that “Results from the negative and positive control probes from the…detection system are shown in Supplementary Figures 2 and 3.”  It would be useful to know more about what is included in these controls.  Do the negative controls lack expressed BRCA1/2 mRNA, or are they simply probe components that lack oligos?

The control probes are designed to either bind to a common housekeeping gene (positive control, UBC) or a bacterial gene (negative control, dapB). We have added this information to the manuscript.

Lines 247-248: “Control probes for housekeeping gene UBC (positive control) tissue and bacterial gene dapB (negative control) were included for each assay.”

5.    In the Supplementary Information, there are a couple of sentences that read like proprietary kit instructions that need to be spelled out as much as possible.  These include line 36, “Pretreat 3 solution was added…” and line 44 “…was incubated with AMP 1-FL for 30 minutes, followed by AMP 2-FL…”

The reviewer is correct in that the protocol is proprietary, and so more detail of what is in each solution, and the probe sequences, is not disclosed by Advanced Cell Diagnostics. This is mentioned in the supplementary methods.

Reviewer 2 Report

This manuscript appears to be an extension of earlier work from the lab to evaluate Potential spliceogenic variants utilizing targeted RNA sequencing. This study utilises RNA in situ hybridisation to measure absolute expression of spliceogenic variants in single lymphoblastoid cells and find that detection of mRNA expression in single cells allows a more comprehensive understanding of how spliceogenic variants influence the expression of mRNA isoforms than RT-PCR. The findings are of interest, but there are a few other issues.

1.     There are 40 BRCA1 and 17 BRCA2 alternate isoforms in LCLs identified by your former research, why you only choose BRCA1c.671-2 A>G and  BRCA2c.7988 A>T in this study?

2.     The biological and clinical significance of the elevated two spliceogenic variants in single cell should be evaluated.

Author Response

1.    There are 40 BRCA1 and 17 BRCA2 alternate isoforms in LCLs identified by your former research, why you only choose BRCA1c.671-2 A>G and BRCA2c.7988 A>T in this study?

As a proof of concept study, two LCLs were chosen to show how RNAscope can be used to detect expression changes of specific mRNA isoforms in single cells. As noted in Lines 231/232, the two LCLs were selected as they contained variants known to disrupt normal splicing patterns.

2.    The biological and clinical significance of the elevated two spliceogenic variants in single cell should be evaluated.

We thank the reviewer for their comment and believe we have discussed the biological significance of our findings in the discussion, particularly Lines 140-162.

Round  2

Reviewer 1 Report

To determine the frequency of exon skipping events in cells carrying spliceogenic mutations or wild type controls, the authors made separate counts of red signals (presumably present in all BRCA1/2 mRNA splice variants) and green signals (present only when there is no skipping of the exon being evaluated).  In many (but not all) cases there were either more reds than greens, or equal numbers of both.  The numbers of signal differences were interpreted as the numbers of exon skipping events.  This description of the analysis was clear in the first draft of the manuscript and was re-emphasized by the text added by the authors in lines 128-130.    

However, the principal concern with the first draft was that no attempt was made to show that probes for different parts of the same molecule gave signals from the same place in the cells.  Figures like Figure 2 and Supplementary Figure 1 show both red signals (presumably present in all BRCA1/2 mRNA splice variants) and green signals (present only when there is no skipping of the exon being evaluated).  One would predict that all molecules emitting a green signal should also emit a red signal when viewed with a different filter.  So the legend to Supplementary Figure 1 is confusing: “A: cellular mRNA that does not carry the targeted deletion as indicated by equal numbers of both green (C1) and red (C2) signals.”  If all green signals visible in these images are masking underlying red signals (instead of merging with them in some identifiable way), then the number of visible greens must be added to the number of visible reds to show the total number of molecules with C2 regions.  Sup fig 1A would therefore show an equal number of deletion molecules and full length molecules, not the suggested lack of exon skipping.  If the green signals and red signals are all coming from separate molecules, then the underlying assumption that all molecules emitting a green signal also emit a red signal has to be wrong.

It should be possible to show two side by side photos of the same cell in the same focal plane with a red filter on one side and a green filter on the other side, demonstrating that the green signals are all in the same positions as a subset of the red signals.  If this is not possible, then the figure legend of Supplementary Figure 1 should be revised to be less confusing.  It could for example say “A: cellular mRNA that does not carry the targeted deletion showing similar numbers of green (C1) and red (C2) signals, indicating a background level of exon skipping. B: cellular mRNA that contains the deletion of interest, with a greater number of red signals (C2) compared to green signals (C1), indicating a higher frequency of exon skipping than seen in wild type cells.”  It should state somewhere in the text what we’re looking at when we see a green signal in an image that shows both red and green signals.  Is it a green signal on top of (masking) a red signal from the same molecule?  Or is it a signal on a molecule that has no corresponding C2 domain?

Author Response

1.    The figure legend of Supplementary Figure 1 should be revised to be less confusing.

We thank the reviewer for this advice and have revised Sup Fig 1 legend to make it clearer.

Supplementary information: “Supplementary Figure 1. Examplar highlighting how the differences in the number of each probe detected in a cell is indicative of the number of mRNA isoforms that carries the targeted deletion. A: cellular mRNA that does not carry the targeted deletion showing similar numbers of both green (C1) and red (C2) signals, indicating a background level of exon skipping. B: cellular mRNA that contains the deletion of interest, with a greater number of red signals (C2; total number of mRNA expressed) compared to green signals (C1; targeting the deletion), indicating a higher frequency of exon skipping than seen in wild type cells.”

2.    It should state somewhere in the text what we’re looking at when we see a green signal in an image that shows both red and green signals.  Is it a green signal on top of (masking) a red signal from the same molecule?  Or is it a signal on a molecule that has no corresponding C2 domain?

The reviewer makes a good point. We mention in the manuscript how each signal was analysed separately under their specific wave length, so eliminating the possibility of one signal masking the presence of another (Lines 260-261). Interestingly, it is possible for the signals from the probes used in this assay to be located on the same molecule up to 2,500 um apart. This suggests that distinct, nonoverlapping signals observed could be located on the same linear molecule, rather than exclusively representing signals on separate molecules. We acknowledge that there are likely other methods of analysis that may have been superior for analysis that could more accurately determine the signals that are located on the same molecule, or on separate molecule, and have added this into the discussion.

Lines 187-188: “Future research may employ confocal microscopy and image analysis tools that enable 3D imaging of transcripts detected using dual probe fluorescence.”